# A New Method for a Polyethersulfone-Based Dopamine-Graphene (xGnP-DA/PES) Nanocomposite Membrane in Low/Ultra-Low Pressure Reverse Osmosis (L/ULPRO) Desalination

**DOI:** 10.3390/membranes10120439

**Published:** 2020-12-18

**Authors:** Lwazi Ndlwana, Mxolisi M. Motsa, Bhekie B. Mamba

**Affiliations:** 1Institute for Nanotechnology and Water Sustainability Research, College of Science, Engineering and Technology, University of South Africa, Florida Science Campus Florida, Johannesburg 1709, South Africa; motsamm@unisa.ac.za; 2School of Materials Science and Engineering, Tianjin Polytechnic University, Tianjin 300387, China; mambabb@unisa.ac.za

**Keywords:** dopamine, desalination, graphene nanoplatelets, mixed-matrix membranes, polyethersulfone, ultra-low-pressure reverse osmosis

## Abstract

Herein we present a two-stage phase inversion method for the preparation of nanocomposite membranes for application in ultra-low-pressure reverse osmosis (ULPRO). The membranes containing DA-stabilized xGnP (xGnP-DA-) were then prepared via dry phase inversion at room temperature, varying the drying time, followed by quenching in water. The membranes were characterized for chemical changes utilizing attenuated total reflectance-Fourier transform infrared spectroscopy (ATR-FTIR) and X-ray photoelectron spectroscopy (XPS). The results indicated the presence of new chemical species and thus, the inclusion of xGnP-DA in the polyethersulfone (PES) membrane matrix. Atomic force microscopy (AFM) showed increasing surface roughness (R_a_) with increased drying time. Scanning electron microscopy (SEM) revealed the cross-sectional morphology of the membranes. Water uptake, porosity and pore size were observed to decrease due to this new synthetic approach. Salt rejection using simulated seawater (containing Na, K, Ca, and Mg salts) was found to be up to stable at <99.99% between 1–8 bars operating pressure. After ten fouling and cleaning cycles, flux recoveries of <99.5% were recorded, while the salt rejection was <99.95%. As such, ULPRO membranes can be successfully prepared through altered phase inversion and used for successful desalination of seawater.

## 1. Introduction

Desalination using reverse osmosis membranes is a costly process due to the high operating pressure required; this drawback currently affects the rate at which various countries adopt the process. As such, many approaches are being investigated to lower these costs, and they include feed water pre-treatment, membrane material modification, and proper plant maintenance. Regarding these RO membranes, it is important to note that these should be developed with favorable properties such as chemical stability, low-fouling, high throughput, high permeability, and low operating pressure [1,2,3].

Low-pressure reverse osmosis (LPRO) membranes were first commercialized in 1995 where the main prerequisites were salt rejection comparable to conventional polyamide types at about 40% less operating pressures [4]. Ultra-low-pressure reverse osmosis (ULPRO) takes this principle a step further by reducing the operating pressures used during desalination. In one study, cellulose acetate/polyethylene glycol-600 membranes were impregnated with Ag particles where the operating pressure was varied between 1 and 5 bars for the mitigation of biofouling. These membranes, obtained via a two-stage phase inversion method were found to possess antibacterial properties against *E. coli* [2]. The ULPRO membranes also allow for the rejection of undissociated organic compounds that can exist in wastewater and brackish water. This property was investigated in a study by Ozaki and Li (2002) where a commercial membrane (ES20) was used [5]. The ULPRO membranes have also found application in the natural gas industry where in one study, both NF and ULPRO commercial membranes (XLE and TFC-S, among others) were investigated for the treatment of produced water extracted from a sandstone aquifer in Eastern Montana, USA, and for iodide recovery [6]. Furthermore, the separation of heavy metal ions (Cu^2+^, Ni^2+^, and Cr^6+^) in the presence of interfering ions (Ca^2+^ and Mg^2+^) has been reported in the literature [7]. In the production of drinking water and/or the recycling and reuse of water, the use of LPRO and ULPRO in terms of modification, performance, stability, and pre-treatment has been reported in other studies [1,3,8,9,10,11,12,13,14,15].

Several steps have been taken by the research fraternity to develop membranes and membrane materials that possess this low-energy-requirement characteristic, as well as enhanced antifouling and mechanical properties. These steps include the modification of materials/polymers and routine preparation methods, both of which can be combined to yield a special kind of membrane so that these two issues can be addressed. As such, nanomaterials have been included in the fabrication of nanocomposite membranes. Nanomaterials such as clays, metal oxides, carbon allotropes (nanotubes, and graphene analogues) have proved useful in the envisaged development of ULPROMs with enhanced properties. Among these, graphene materials have amassed expansive interest due to their high Young’s modulus (for tensile strength). The mechanical properties of graphene surpass those of the previously mentioned materials. Graphene possesses a one-atom-thick two-dimensional sheet structure of *sp^2^*-hybridized carbon atoms [16,17]. Properties of graphene also include great flexibility and a small interlayer spacing (ca. 0.34 nm). This interlayer spacing is critical as it allows for enhanced cation/anion separation and water transport properties [18]. The further intrinsic impermeability and selectivity of graphene allow for the passage of water molecules while sieving unwanted hydrated metal ions. Additionally, the hydrophobic nanochannels of graphene result in improved water transport properties [19,20]. However, disadvantages of the structural conformation of graphene is such that it is destabilized in water, and usually mitigated by the inclusion of multivalent metal ions [21]. Additionally, surface graphene-modified polyamide thin film nanocomposites can be degraded by its oxidation in chlorine-containing water during RO, and further exacerbated by the presence of metal ions [17]. These said properties have been improved by stabilizing/coating graphene with monomers or polymerized/crosslinked networks of tannic acid (TA) or dopamine (DA). These modification approaches allow for the fine-tuning and control of the interlayer spacing of graphene, and consequently, the preparation of membranes with improved mechanochemical resistance, water permeation fluxes, salt separation, and antifouling properties [20,22,23,24].

With this literature, there is a window of improvement in terms of the modification and membrane fabrication methods to yield further enhanced properties as previously discussed. Other works have presented the use of dry/wet phase inversion, mostly for gas separation as per literature where pristine polymers in casting solutions have been utilized [25,26]. It would be greatly interesting if such methods could be applied and augmented to yield ultra-low-pressure reverse osmosis membranes for seawater desalination applications.

This work presents a simple and unique modification and fabrication method for the preparation of dopamine-stabilized graphene-based PES nanocomposite membranes. The xGnP^®^ (nanosized graphite flakes [27]) were chemically stabilized with DA and incorporated into the PES casting solution which was used to prepare the ULPRO membranes. The newly fabricated mixed matrix membranes (MMMs) were thereafter characterized using ATR-FTIR, XPS, SEM, AFM, and other instrumental techniques, and then applied in the ULPRO process using synthetic seawater streams.

## 2. Materials and Methods 

### 2.1. Materials

Polyethersulfone 3100P (PES powder, *M*_w_ = 31–58 kDa) was obtained from Solvay (Beveren, Belgium), graphene nanoplatelets (xGnP) with a surface area of 300 m^2^, *N*-methyl-2-pyrollidone (NMP, 99.5%), dopamine hydrochloride (DA, 98%), sodium dodecyl sulphate (SDS, 99.0%), and polyvinyl-2-pyrollidone (PVP, 98%), were purchased from Sigma-Aldrich (Kempton Park, South Africa). The NaOH pellets (98%), ethanol (≤99.5), and HCl (37%) were obtained from Merck (Germiston, South Africa). Water with a resistivity of 18.2 MΩ.cm was utilized for all the experiments and was produced by a Millipore reverse osmosis system (Merck Millipore, Tokyo, Japan). 

### 2.2. Methods

#### 2.2.1. Preparation of DA-Stabilized xGnP (xGnP-DA)

The xGnP-DA suspension was prepared by sonication of the pristine xGnP in a mixture of DA and NMP as in our previous work and used further as is [18]. The synthetic route involved weighing pristine xGnP^®^ (120 mg) and 1g DA into 9 mL of NMP and dispersing the mixture for 4 h under ultrasonication at 45 °C.

#### 2.2.2. Membrane Fabrication

The casting solutions were prepared by adding 1 mL of the xGnP-DA suspension to the casting solution which consisted of 18% PES and 0.5% PVP in NMP. The result was a 1 m/m% loading of the xGnP-DA suspension in the casting solution. The composition and chemistry of the xGnP-DA is known from our previously reported work [18]. The resulting casting solution was then subjected to 2 h of ultrasonication at 45 °C for full dispersion. The membranes were fabricated by drawing down the casting solutions with a doctor blade set at an air gap of 200 µm on a clean glass plate. The glass plates were then placed under a fume hood to evaporate the solvent without forced convective evaporation or induced humidity. This process was carried out at curing times varying between 4 and 48 h, as the film dries and cures at room temperature. After this elapsed period, the casted film was quenched in water as a second step of phase inversion, allowing for membrane formation. The fabricated membranes were then designated as tabulated in Table 1.

### 2.3. Characterization

#### 2.3.1. Surface Chemistry

For the determination of surface chemical changes, all the membrane samples were analyzed on a Spectrum 100 ATR-FTIR system was used (PerkinElmer Inc., Waltham, MA, USA). The probing range set for the instrument was from 650–3700 cm^−1^ with an operating resolution of 4 cm^−1^. All the samples were dried in a desiccator for 24 h before analysis. The ATR-FTIR analysis was confirmed using the multiprobe XPS (Omicron NanoTechnology GmbH, Taunusstein, Germany) for the chemical bonding and compositional changes. The instrument was calibrated to the C 1s peak at 284.6 eV. The core levels of interest were the C 1s, N 1s and O 1s together with their binding energies (B.E). For this purpose, only the P04h and X48h membranes were analyzed as the pristine and highest permutations of the chosen synthetic method.

#### 2.3.2. Surface Morphology

Changes in surface morphology for the prepared membranes were monitored using a JSM-IT300 scanning electron microscope (JEOL, Tokyo, Japan). To achieve this, the membranes were cut into thin strips and gold-coated using the Q150R ES sputter coater (Quorum Technologies, London, UK). SEM cross-sectional analysis of the internal morphology of the membranes was also carried out. To obtain cross-sections of the membranes, the specimens were dipped into liquid nitrogen and subsequently fractured to reveal the internal structure for analysis. A Nanoscope IV SPM Controller as the atomic force microscope (AFM, Veeco, Plainview, NY, USA) was used to analyze surface topography and thus, measure surface roughness (i.e., arithmetic mean roughness (*R_a_*)). The membrane pieces (1 cm × 1 cm) were mounted on the nickel stubs and placed onto the stage and examined using a Si cantilever. The AFM noncontact mode was used in air for all samples.

#### 2.3.3. Membrane Water Contact Angle

Contact angle measurements were performed to determine the surface hydrophilicity (wettability) of the membranes. Drop shape analysis (DSA) was used to determine the contact angles of the membranes using a DSA 100 goniometer (Krüss GmBH, Hamburg, Germany) to investigate the extent to which water interacts with the surfaces. The membrane samples (three each) were placed onto the sample stage and measurements of sessile drops taken on more than ten positions and averaged.

#### 2.3.4. Pure Water Permeation

Pure water permeation is another critical membrane parameter. Measurements were taken using an HP4750 stainless stirred dead-end cell (Sterlitech Corp., Kent, WA, USA) fed with a 10 L pressure vessel. The pressure vessel maintained and provided a steady supply of nitrogen gas for improved control of the operating pressure to the cells between 1–8 bars. The membranes (in triplicate) used were circular with 1.48 × 10^−3^ m^2^ as the effective area. The pure water fluxes were calculated according to the Equation (1):(1)$$J_{w}\;=\frac{Q}{A\cdot t}$$
where J denotes the water flux (L/m^2^ h), Q stands for the permeate volume, t is the time interval over which the permeate is collected, while P denotes the operating (transmembrane) pressure, and A is the membrane’s effective area (m^2^).

#### 2.3.5. Water Uptake, Porosity, and Pore Size

Water uptake, porosity, and pore size were also determined for the prepared membranes (three from each membrane coupon), as these parameters are interlinked. Gravimetric analysis is used to measure these parameters, down to the membrane pore size which was determined using the Guerout−Elford−Ferry equation [28]. In the cited work, the authors investigated both nano and sub-nanosized pores using this formula as shown in Equation (2):(2)$$r_{m}\;=\sqrt{\frac{(2.9-1.75\varepsilon )8\eta lQ^{\;\;}}{\varepsilon AP}}$$

The methods are detailed further in our previous work [18].

#### 2.3.6. Desalination Using NaCl and Synthetic Seawater Solutions

To interrogate the desalination efficiency of the membranes prepared in this work, the desalination of NaCl (3000 ppm) and synthetic seawater solutions were used. The simulated seawater used in this study was prepared to have the composition as indicated by Liao et al. (2020) [29] and depicted in Table 2.

The physiological make-up of the simulated seawater was as tabled in Table 3.

The feed and permeate were measured for conductivity to determine the salt rejection, R which was calculated using Equation (3):(3)$$R\%\;=\;\left.\left(1-\;\frac{C_{p}}{C_{b}}\right.\right)\times 100$$
where $C_{p}$ is the concentration of the permeate and $C_{b}$ is that of the retentate.

With these salt rejection experiments, salt permeation fluxes were also taken similarly to Equation (1).

The salt rejection measurements were taken at increasing operating pressures to investigate the possible trade-off effects between the solute retentions and increasing permeate fluxes. One membrane (in triplicate) was used for each the salt rejection experiments at different pressure values from 1 to 8 bars.

The physicochemical characteristics of the water were also determined before and after salt rejection where these were the conductivity, total dissolved solids (TDS) (CON 700, Eutech Instruments, Singapore). This specific instrument is extremely sensitive with the ability to measure extremely low to high amounts of solids in solution (from virtually undetectable to 100 ppt/200 mS). As such, these parameters can be reliably measured with this instrument. Additionally, pH measurements of the solutions were measured using the pH 80+DHS STIRRER (XS Instruments, Carpi, Italy). The lowest levels of these three parameters indicated the quality of the water and the effectiveness of the membranes in envisaged seawater desalination.

## 3. Results

### 3.1. Surface Chemistry of the Prepared Membranes

The membranes were then fabricated from the casting solutions and Figure 1 presents their ATR-FTIR spectra. The pristine membrane (P04h) showed the intrinsic PES bands and peaks between 1378–1578 cm^−1^, attributed to its aromatic vibrations and stretches. After the inclusion of xGnP-DA, several peaks and bands intensify such as the aromatic C=C–H functional groups at 3070–3096 and 1679 cm^−1^, especially notable for the X24h membrane. These peaks indicate the physicochemical interactions between PES and xGnP-DA, and further the π–π stacking between the aromatic functional groups of PES and xGnP-DA. The appearance of the broad peak around 3440 cm^−1^ showed the presence of hydrogen-bonded amine groups of DA through their interaction with the carboxylic groups from the xGnP. However, with further drying time, the X48h membrane indicated decreased intensities of these peaks and bands, and there was a shift in the aromatic band from 1678 cm^−1^ (X24h) to 1682 cm^−1^. This band shift and reduction in intensity can be attributed to the extended time duration where the interactions between xGNP-DA with PES (through π–π stacking) were intensified. The detection of these functional groups and their bands/peaks has been reported in the literature for membranes prepared from graphene, DA, and PES [30,31,32,33,34].

X-ray photoelectron analysis (XPS) was also carried out on the membranes prepared to corroborate the data obtained on ATR-FTIR and confirm chemical bonding. Furthermore, surface coverage was also measured for the various chemical species before and after modification at core levels C 1s, N 1s, and O 1s. Table 4 presents the changes in these parameters to the modification and fabrication method. Figure 2 depicts the XPS spectra for the deconvoluted peaks. Figure 2a presents the C1s peak for the PES-air membrane which could be deconvoluted to several species, namely C–C, C=C, C–H (284.6 eV), and C–O (285.6 eV) which constituted 50.3% and 19.4%, respectively. It was also observed that C–N amounted to 30.3% surface coverage, which could be deemed an anomaly as the hydrophilic segments of pore former PVP traversed to the membrane surface. This led to a disproportionate amount of C–N species quantification as per Table 4. The disproportionate amounts of hydrophilic segments were observed to be typical for PES as previously reported in another work [35]. Upon modification of the casting solutions and the subsequent fabrication of the MMMs, there was a reduction in the aliphatic/saturated species (to 40.3%) where these new peaks were observed to appear as shown in Figure 2b on the C 1s spectrum. The C–O species were observed to increase from 19.4% to 50.4%, indicating that the xGnP-DA contributed these functional groups to the modified membranes. The modification also resulted in the reduction of C–N (9.8%) components as contributed to the membrane matrix by DA where they appeared at the binding energies of 287.2 eV. New components were also detected such as the C–OOH at 288.7 eV and surface coverages, respectively, measured at 2.8% [16,36,37]. This low level of C–OOH detection could be attributed to the reduction of xGnP by DA during the stabilization (cross-linking) reaction. This crosslinking could also explain the exceptionally low detection of C–OOH as their contribution from xGnP was reduced due to the chemical bonding with DA, increasing the presence of C–O species of the fabricated membranes.

In addition to the abovementioned chemical changes, nitrogenous species were also observed, as previously alluded to by ATR-FTIR and the analysis in question. These nitrogenous components such as the N–H and –NH_2_ appeared at 398.4 and 399.9 eV, respectively (Figure 2d). This observation suggests that the DA was successfully chemically included in the xGnP-DA molecular structure within the membrane matrix. Furthermore, it was recorded that there was only one oxygen species before modification as can be seen in Figure 2c, which upon modification, increased in width as per Figure 2d. Furthermore, there were changes to the oxygen species as compared in Figure 2e,f. As can be seen, the PES-air membrane only possessed O–C– species, and these were attributed to the ether functional groups of the pristine PES. However, on modification with xGNP-DA, new alcoholic and carboxylic species were introduced to the xGnP-DA/PES-air matrix. This indicated the formation of a new oxygen-bonding species of C–OOH where this functional group was observed on ATR-FTIR to confirm its successful incorporation into the xGnP-DA/PES-air matrix. Other nanocomposites and membranes have been characterized and chemical functionalities similar to these observed in this work confirmed by XPS [23,31,38].

### 3.2. Surface Morphology

Figure 3 depicts the SEM images (with inserts at 4500× magnification) taken for the membranes as fabricated in this study. The morphology of the pristine PES membrane after 4 h of ambient drying time (P04h) and the subsequent quenching in water is indicated in Figure 3a. The microstructure of the membrane in question was that of an open and porous nature with a thin skin layer (Figure 3a’). The sublayer presented a spongy structure that morphed to large macrovoids towards the bottom of the membrane. This morphology was influenced by both the drying time and the quenching in water (dry/wet phase inversion). This kind of structure was distinct from that of pristine PES membranes as prepared via wet phase inversion in water as reported in the literature [32,35,39,40]. The pristine PES membranes in the cited studies indicated slanting pores that extended to the bottom of the membrane because of the thermodynamics of demixing between solvent and nonsolvent. Furthermore, due to the extended periods of evaporation, the P04h membrane was also more compact in terms of thickness (103 µm). Upon modification, performed under identical evaporation periods, the X04h membrane presented the formation of a denser skin layer (Figure 3b). This skin layer as depicted in Figure 3b’, occurred due to the presence of xGnP-DA within the membrane matrix. The xGnP-DA could be observed as particles distributed throughout the membrane matrix, which were absent in the pristine membrane. Furthermore, the measured thickness of this membrane was 80.0 µm, and the discrepancy between this value and that of P04h was attributed to the presence of xGnP-DA. With increasing drying time, the skin layer also became more prominent on the X08h membrane where its measured thickness was 158 µm (Figure 3c). Again, it was the presence of xGnP-DA which slowed down the loss of solvent at this drying time of 24 h. This resulted in a spongy structure with semi-finger-like macrovoids with a membrane thickness of 156 µm and dense skin layer (Figure 3d,d’). The negligible loss in membrane thickness could be credited to the rate of solvent evaporation slowing down with drying time. The finger-like pore structure became even more pronounced for the X48h membrane, with the membrane thickness being measured to be 102 µm because of the prolonged period of drying (Figure 3e). It was noted that the xGnP-DA/PES casted film behaved exceptionally differently with extended periods of drying. The xGnP-DA particulate material was then pronounced in visibility on the thick skin layer as a direct consequence of large amounts of solvent evaporating. Additionally, this increased presence of xGnP-DA on the membrane surface could be ascribed to its water-compatible nature which caused its upward trajectory during the wet phase inversion stage (Figure 3e’). The thickness of the skin layer was critical due to its ability to limit solute transport, to attain membranes which can be applicable for seawater desalination applications. Furthermore, the presence of graphene within this network also enhanced the size exclusion of anions and cations due to its small interlayer spacing (~0.34 nm) [30,33,41,42,43].

Figure 4 illustrates the surface analysis of the membranes by both scanning and atomic force microscopy. The SEM images were placed alongside their AFM micrographs to interrogate the membrane surface features. As can be observed in Figure 4a1–c1 that the SEM and AFM analysis presented identical features for the P04h membrane due to its pristine nature. It can be observed that this membrane possessed a porous filtration layer, with large depressions and visible pores. It should be noted that these depressions are not pores as the latter are visible through the former. These depressions occur due to the four-hour drying period interval as the solvent evaporates at ambient and uninterrupted conditions, and the subsequent quenching which causes large pores to form through these depressions. A few seconds of drying time, or none does not result in the kind of morphology as observed for this membrane, as previously reported in the literature [40,44,45]. The large pores, as observed on the said micrographs, also supported the SEM cross-sectional analysis; and this was influenced by the rapid demixing rates of the polymer phase in water during the quenching step of phase inversion. The surface roughness measurement as recorded for this membrane was 58.611 nm. The surface roughness dictates water-membrane interactions and the resulting fouling propensity [6,46]. Consequently, due to the chosen modification and fabrication methods, an increasing size in the depressions was observed on both SEM and AFM, with the pores on the surface, and yet again, visible through the depressions. As such, increasing surface roughness measurements were recorded when compared to the pristine membrane. This is attributed to the presence of xGnP-DA within the membrane matrix and thus, the surface of X04h. It was also visible that the pore size had increased from that of the pristine. With increasing drying time, the X24h membrane presented a lower R_a_ value of 37.673 nm due to the smoothing of the membrane surface. This observation can be credited to the slow migration hydrophilic segments during the longer periods of solvent evaporation. However, after 48 h of drying, the X48h membrane presented an even rougher surface, due to the heightened presence of xGnP-DA/PES segments towards and on the membrane surface. These segments showed worm-like structures which increased in entanglement with increased drying time throughout the modified membranes. These can be seen in the higher resolution images in Appendix A. As a result, increased surface roughness measurements of 85.025 nm were recorded for this membrane. It can be noted, more importantly, that there were no pores or depressions visible for this membrane as compared to the P04h and X04h permutations. This was attributed to the increased amounts of solvent that were lost during the extended periods of drying, exposing high concentrations of xGnP-DA to the surface of the membrane. This, in turn, resulted in restricting pore formation, but an increase in the surface roughness. These results further confirmed the presence of xGnP-DA and the surface modification as previously demonstrated by FTIR, XPS, and SEM cross-sectional analyses. The surface roughness values recorded for these nanocomposite membranes were observed to be higher than that of the pristine (P04h). The surface roughness was observed to increase due to the modification because of the hydrophilic segments present on the substrate, in agreement with the literature [33].

### 3.3. Membrane Water Uptake and Porosity

Water uptake and porosity measurements were performed for the membranes prepared in this study. Figure 5 presents this data where the pristine membrane (P04h) indicated a water uptake of 301% relative to its dry state. Upon modification, this parameter dropped sharply as can be observed for X04h. This value was then observed to increase with drying time (for the X08h membrane) and this observation was attributed to the equilibration of the macrovoid spaces with increased drying time. Nevertheless, it decreased again for the X48h membrane and this was credited to the compaction of the membranes with drying time as was observed on the SEM analysis. The results of the water uptake experiments were subsequently utilized to calculate membrane porosity. The P04h membrane presented a porosity of 85.9%, a result corroborative of the SEM and AFM analysis. Upon modification with xGnP-DA, the porosity presented a decline to 60.5% as the modifier reduced the formation of numerous pores even though they appeared larger. However, the drying time proved to be a factor regarding porosity as the loss of solvent increased the viscosity of the cast film. As such, the porosity was reduced, and the further increase in drying time increased the porosity because the water-compatible hydrophilic segments of xGnP-DA traversed towards the membrane surface. This, as a result, increased the rate of numerous small pores on the membrane surface of X48h as per AFM results. It is also important to note that the presence of xGnP-DA for this membrane increased water uptake due to the disproportionate presence of this hydrophilic modifier existing towards the skin layer. Membrane swelling is a property reported to increase permeability, and thus, results in lower operating pressures. As such, this factor can be important in the development of low energy ULPRO membranes for desalination applications [14]. However, extensive swelling can result in the membrane losing both chemical and mechanical integrity through the destabilization of the polymer chains. As such, the modification and fabrication methods resulted in significantly reduced membrane swelling by 53.5%. Reduced swelling is known to also affect the conformation of graphene sheets as they are water-swellable. This swelling can result in the increased interlayer spacing of graphene, and thus decreased efficacy during the rejection of inorganic anions and cations [47].

### 3.4. Pore Size and Contact Angle of the Fabricated Membranes

Pore size determination was also carried out for the membranes prepared as presented in Table 5. The membrane, P04h possessed an average pore size of 0.21 µm, and this was due to the pristine nature of the PES casting solution. There was a decrease in pore size for the X04h membrane to 0.17 µm, consistent with the observations made during the SEM and AFM analysis. Nevertheless, pore size was observed to further decrease until X48h, and this was attributed to the evaporation of solvent during the drying step of phase inversion. The prolonged period intervals of solvent evaporation encouraged an increase in the viscosity of the cased film. As a direct consequence, the compaction of polymer chains ensued, which in turn, tightened the membrane pores. As a result, the recorded pore size of the X48h membrane was 0.28 nm, and thus falls in the realm of RO filtration. This means that this specific membrane can be used in the desalination of seawater to yield pure and drinkable or reusable water. In terms of contact angle, the membranes prepared did not present a greatly significant change as a response to the modification and fabrication method. The P04h membrane possessed a contact angle of 66.2°, and this is attributed to the large pore size and high porosity. The X04h membrane presented a contact angle of 68.1°, while increasing the drying time to 48 h resulted in that of the final X48h membrane presenting a reading of 67.5°. This means that the membranes possessed a decent degree of hydrophilicity or wettability. Hydrophilic membrane−water interactions are a critical parameter in maintaining membrane productivity by reducing the likelihood of fouling. Fouling is an undesired phenomenon as it reduces water permeation fluxes (through pore blockage) and this increases energy requirements and cleaning costs [45,48,49].

### 3.5. Dynamic Pure Water Permeation Studies

Studies on pure water flux as a function of operating pressure (1–8 bars) were also carried out to determine the water transport properties of the newly developed membranes. It can be observed that the P04h membrane possessed the highest fluxes across the pressure range (Figure 6). This was anticipated due to the open and porous structure as evidenced by the high porosity and water uptake by this membrane as per the previous discussions. With modification, the X04h membrane presented a sharp decrease in water permeation fluxes, and this was due to the presence of xGnP-DA within the membrane matrix. The combination of the slow phase change and the presence of xGnP-DA may have led to the formation of a compact structure that resisted the diffusion of water. This trend was observed until the final membrane in this series, X48h. However, with pressure, the pure water permeate fluxes were observed to increase as a direct response to the force of water flow through the membrane. The insert in Figure 6 depicts the permeation response of the X48h membrane to increasing pressure. The values recorded for this specific membrane were calculated to be between 1.68 and 19.00 L/m^2^ h with increasing operating pressure. In membrane applications, water permeation fluxes are a key property in the quest to retain high throughput as has been reported in the literature surveyed [9,16,50,51].

### 3.6. Desalination of Synthetic Seawater and Salt Permeation Fluxes

Salt rejection studies were also carried out to assess the efficiency of the membranes for potential RO applications. These experiments were carried out between 1 and 8 bars as illustrated in Figure 7. Initially, NaCl rejection experiments were carried out as a preliminary study. As indicative of the observations in Figure 7a (3000 ppm NaCl), at the operating pressure of 1 bar while rejecting <99.99% of the monovalent salt. However, there was a decreasing trend in the rejection of this salt, but only by an insignificant decline of less than 0.1% between 1–8 bars operating pressure. After these observed, unanticipatedly high NaCl rejections, synthetic seawater desalination experiments were then carried out to further interrogate the selectivity of the new membranes regarding the desalination of synthetic seawater (Figure 7b). As such, because of the pore sizes presented in the previous section, the results indicated that the membranes (P04h–X24h) were between microfiltration and ultrafiltration. The consequence of this was the observation that makes such membranes inapplicable for desalination applications, and thus, it was only the X48h membrane that was used for the salt rejection studies. It can be observed that this membrane was able to reject <99.99% (at 1 bar operating pressure) of the anions and cations in the synthetic seawater solution. These dead-end salt rejections were also observed to be higher than those reported in the literature and some of their pristine and modified commercial RO counterparts [52,53]. This can be attributed to the small pore size and the interspacing of graphene which enhance the size exclusion of anions and hydrated cations, producing pure water. Furthermore, the Donnan effects are increased as imparted by the xGNP-DA where the steric effects can also be credited with these high rejections [22,52]. These salt rejections remained above 99.99% with increasing operating pressure up to 8 bars. These observations were made even though there was a twenty-fold increase in the recorded fluxes (1.27 to 25.89 L/m^2^ h) between the operating pressures of 1 bar to 8 bars. As such, the insignificant effect of operating pressures (and thus, permeate fluxes) on salt rejections were in stark contrast to most works in the literature, suggesting enhanced water permeability and salt selectivity for the newly fabricated membranes [53]. In addition to these observations, operational fundamentals of U/LPRO membranes require variation in working pressures to mitigate fouling which can reduce the operational lifespan of the membranes [54]. The ability of this membrane to maintain high salt rejection even with increasing operating pressures means that the switching of the latter parameter is possible; thus, increasing the scope of potential desalination applications. This can be attributed to the modification and fabrication method which resulted in enhanced filtration properties imparted by xGnP-DA within the membrane matrix. It is the reduced pore size, together with the chemical functionalities (COOH, CONH, NH_2_) of the xGnP-DA that allow for augmented electrostatic exclusion of charged species (Mg^2+^, Ca^2+^, SO_4_
^2−^, Cl^−^, Na^+^, and K^+^) over the (pDA) coating approach as applied in other works [24,30,34,42]. The operating pressure also affected salt permeation fluxes as marked in blue on the figure in question. Salt permeation fluxes provide deeper insight into real-world water transport properties of the membranes during desalination. This value was recorded to increase with pressure from 1.27 L/m^2^ h at the operating pressure of 1 bar to 26.89 L/m^2^·h at 8 bars. This was the observed tradeoff between salt rejection and salt permeate fluxes where the former was high and the latter, low as per other works in the literature [34].

### 3.7. Effect of Fouling Exposure and Cleaning

The membranes prepared were then exposed to SDS as a model foulant to interrogate antifouling, and subsequently their performance after cleaning. Membrane−foulant interactions were also investigated by the use of ATR-FTIR. Figure 8 presents the recorded data where it illustrates the resulting FT-IR spectra. The spectrum of the foulant, SDS revealed C–H stretches between 2849 and 2954 cm^–1^ with an –SO_2_ band also observed at 1217 cm^−1^. Upon the P04h membrane being exposed to SDS, there was an intensification of the alkyl stretches between 2928 and 2993 cm^−1^. The blue shift of these peaks from those of SDS, and a red shift and intensification of the aromatic band from 1666 to 1676 cm^−1^ may be indicative of the interactions between the PES phenyl ring with alkyl groups of SDS. However, this means that there was an adsorption of SDS onto the membranes, but the interactions were not of a chemical nature. This opinion is further informed by the lack of sulfonic groups that would appear around 1200–1400 cm^−1^ for the fouled X48h membrane. This can be attributed to the presence of xGnP-DA which discourages extensive adsorption of SDS. Furthermore, the ease of washing with H_2_O_2_ where the X48h membrane showed a similar spectrum prior to fouling with SDS, with the further attenuation of the alkyl stretches as contributed to by the foulant as previously noted. As a result, the X48h membrane retained an almost identical spectrum after washing relative to its unused state (see Figure 1). Contact angle measurements were also carried out (as per Figure 8b), and these revealed that the washing with hydrogen peroxide resulted in a slight reduction in this parameter throughout the fouling and washing cycles. This is easily explained by the fact that the acidic nature of H_2_O_2_ removes the foulant and may subsequently protonate the functional groups of the membrane (amine and carboxylic). As such, this increases the membrane−water interactions as its water sorption increases from the values recorded pre-fouling exposure. This occurred without significantly altering the surface morphology of the membrane as illustrated in the SEM analysis and Figure 8b [55]. This observation was evident by the contact angle reducing from 67.5° to 64.3° after ten fouling and washing cycles.

Flux recovery ratio (FRR) and salt rejection loss studies after fouling and cleaning were carried out and the data is presented in Figure 8c. The rejection studies indicated that there was no significant loss in this parameter as the recorded values remained above 99.9% with increasing fouling and cleaning cycles. This was particularly impacted by the membrane cleaning protocol and decreasing contact angle. Together with this result, the membrane retained up to 99.52% of the initial pure water permeation fluxes. These combined results mean that the membrane can be cleaned without severely impacting the imparted properties. Furthermore, this implies that the membrane possesses high reversible fouling (and low irreversible fouling), parameters used as a measure of antifouling properties. Consequently, the membrane developed in this work through this unique method, possesses promising properties in terms of salt rejections, flux recoveries, and thus it is a candidate for ULPRO applications. These results present a significant improvement to existing literature where the best flux recoveries were the relationship between the adsorption of foulant adhesion resulting in the reduction of pure water permeate fluxes. As such, their observations led to lower flux recoveries due to the chosen modification approaches as compared to this work [15,24,49,54,56].

## 4. Conclusions

This work envisaged the development of a new fabrication method and the development of ultra-low-pressure reverse osmosis (ULPRO) membranes for simulated seawater desalination. Conventionally, two-stage phase inversion is used to prepare gas filtration membranes, and this study reports on the preparation of ULPRO membranes using this approach. As achieved, the facile method combined dry phase inversion coupled with quenching in water. The membrane fabrication method presented did not use forced convection evaporation, induced humidity, or heating. The physical properties in terms of contact angle, water uptake/swelling, porosity, and pore size were affected by the prolonged periods of drying/solvent evaporation. The direct and most dramatic consequence of drying time was seen in the pore size reduction from micro to sub-nanometer (X48h) size, and thus these could be investigated for RO to ULPRO applications. This was due to the salt rejection abilities not being greatly affected by increasing operational pressures from 1–8 bars, with the rejection ranging at <99.99% and decreasing to <99.99 which are higher than those reported in the literature. Pure water permeation fluxes proved to be tunable with operating pressure, with negligible negative effects on salt rejection. The membrane proved to resist fouling for up to ten filtration and cleaning cycles, maintain high flux recoveries (99.9%), and salt rejections (99.95%) after cleaning. The method of modification and fabrication of this membrane, as a result, shows great promise in the development of sustainable methods for nanocomposite membranes in low energy/cost U/LPRO desalination.

## Figures and Tables

**Figure 1 membranes-10-00439-f001:**
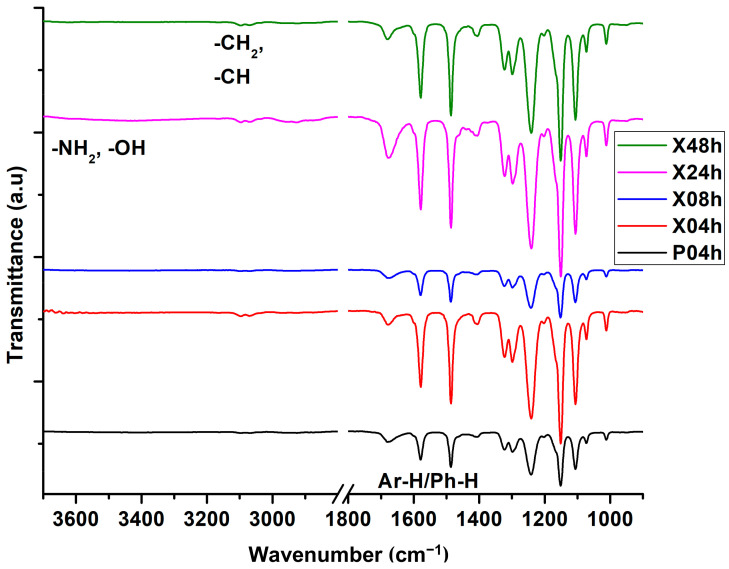
Attenuated total reflectance (ATR)-FTIR spectra of the pristine membrane (P04h) and of the newly prepared membranes (X04h; X08h; X24h; and X48h).

**Figure 2 membranes-10-00439-f002:**
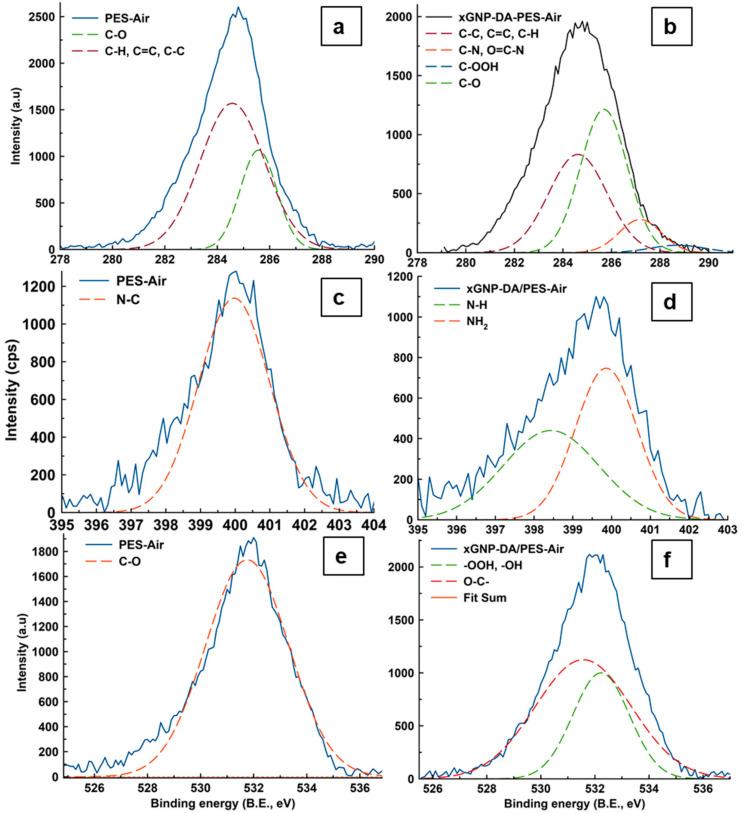
The XPS deconvoluted spectra for the respective pristine (P04h) and modified (X48h) membranes for core levels where (**a**) and (**b**) represent C 1s; (**c**) and (**d**), N 1s; with O 1s shown on (**e**) and (**f**).

**Figure 3 membranes-10-00439-f003:**
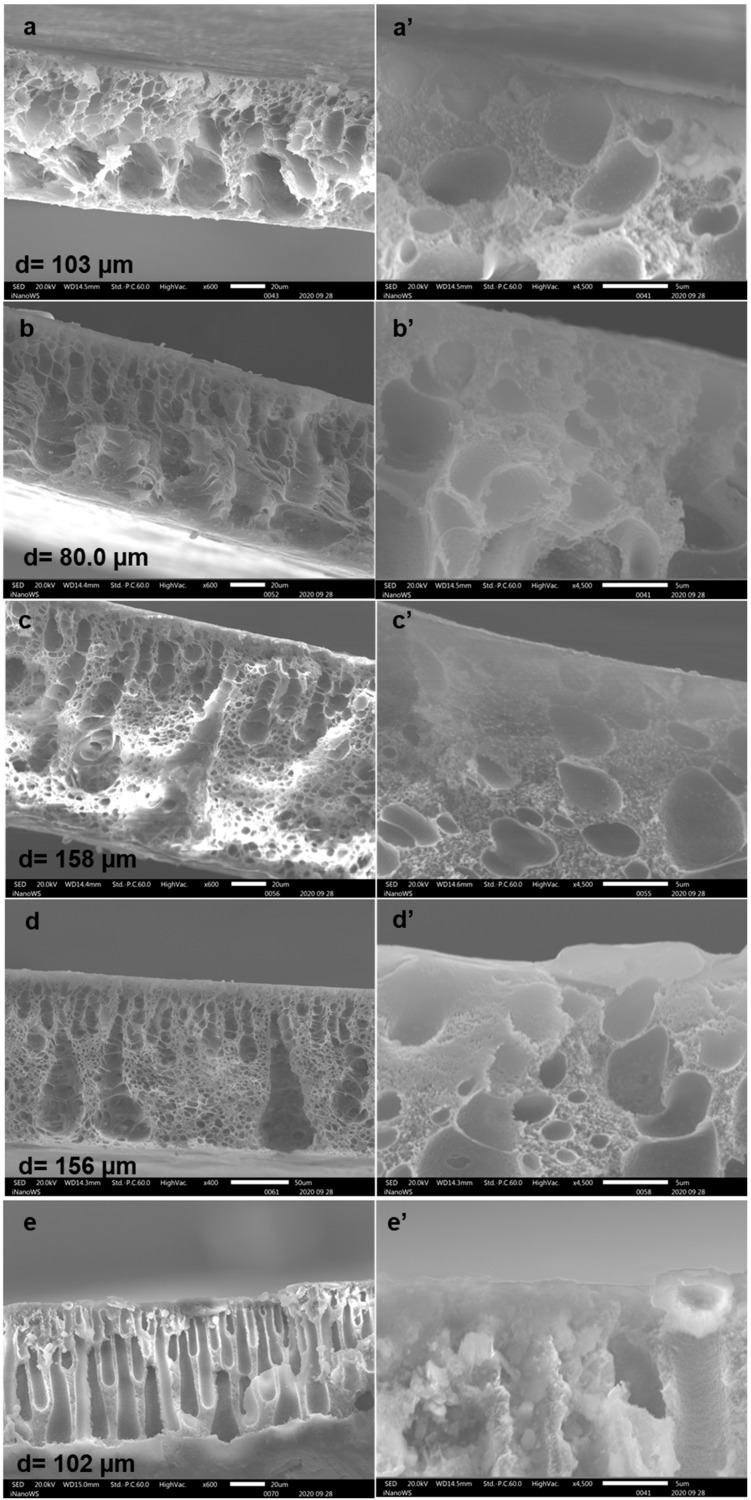
SEM micrographs for the membranes prepared: (**a**) pristine PES (P04h), (**b**) X04h, (**c**) X08h, (**d**) X24h, and (**e**) X48h at 400× magnification, respectively where (**a’**–**e’**) indicate the respective skin layer formation of the membranes 4500×.

**Figure 4 membranes-10-00439-f004:**
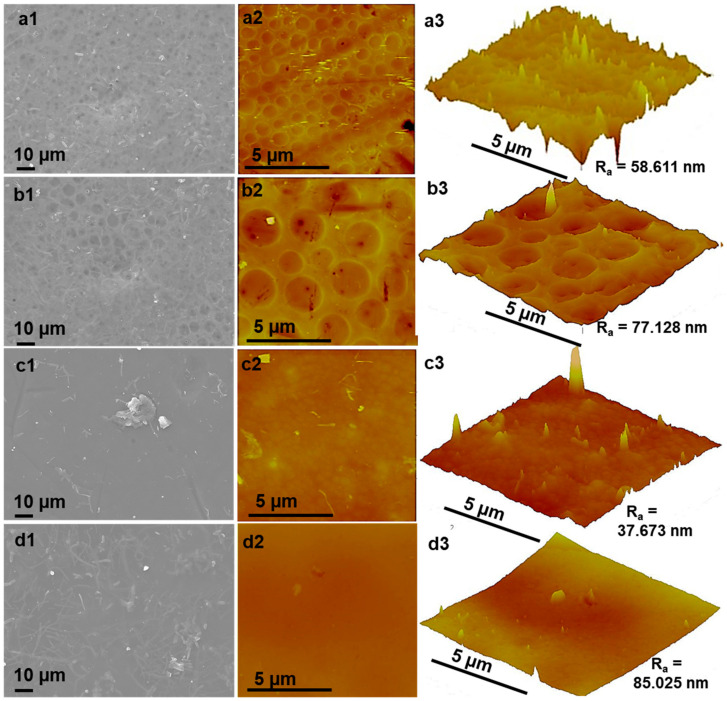
Superficial analysis of the membranes prepared: (**a**) P04h, (**b**) X04h, (**c**) X24h, and (**d**) X48h, where (**a1**–**d1**) are SEM images at 1000× magnification, (**a2**–**d2**) are the topological AFM images, and (**a3**–**d3**) are topographical AFM micrographs at 5 µm × 5 µm which indicate the surface roughness measurements (R_a_).

**Figure 5 membranes-10-00439-f005:**
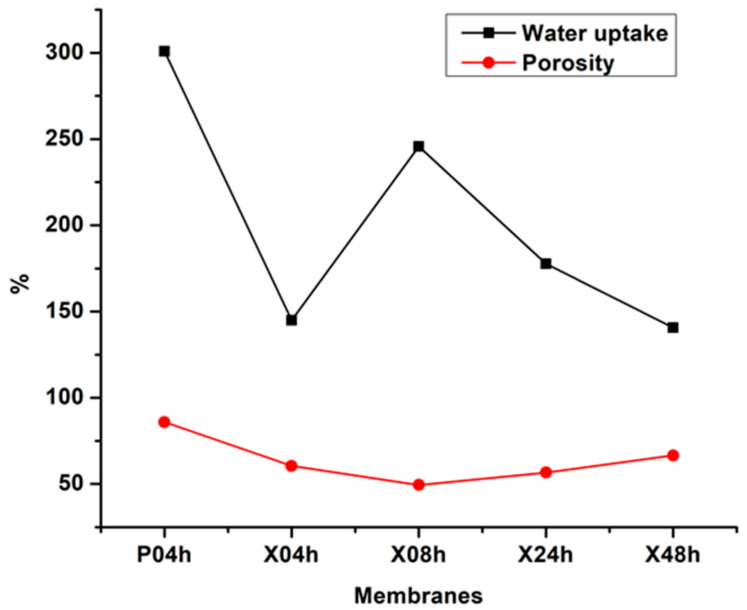
The water uptake and porosity measurements for the membranes P04h, X04h, X08h, X24h, and X48h.

**Figure 6 membranes-10-00439-f006:**
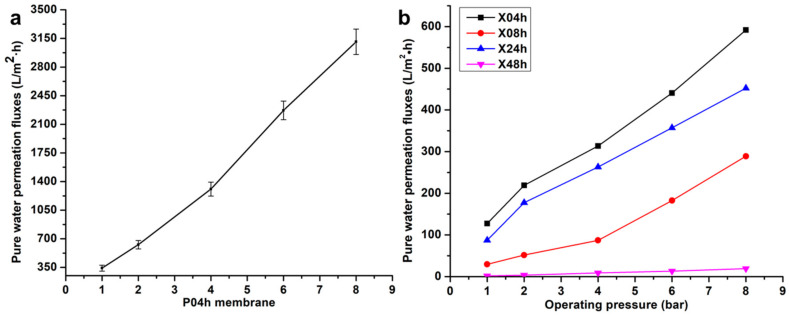
Dynamic pure water permeation fluxes for the membranes prepared: (**a**) the P04h membrane, and (**b**) the modified membranes (X04h, X08h, X24h, and X48h).

**Figure 7 membranes-10-00439-f007:**
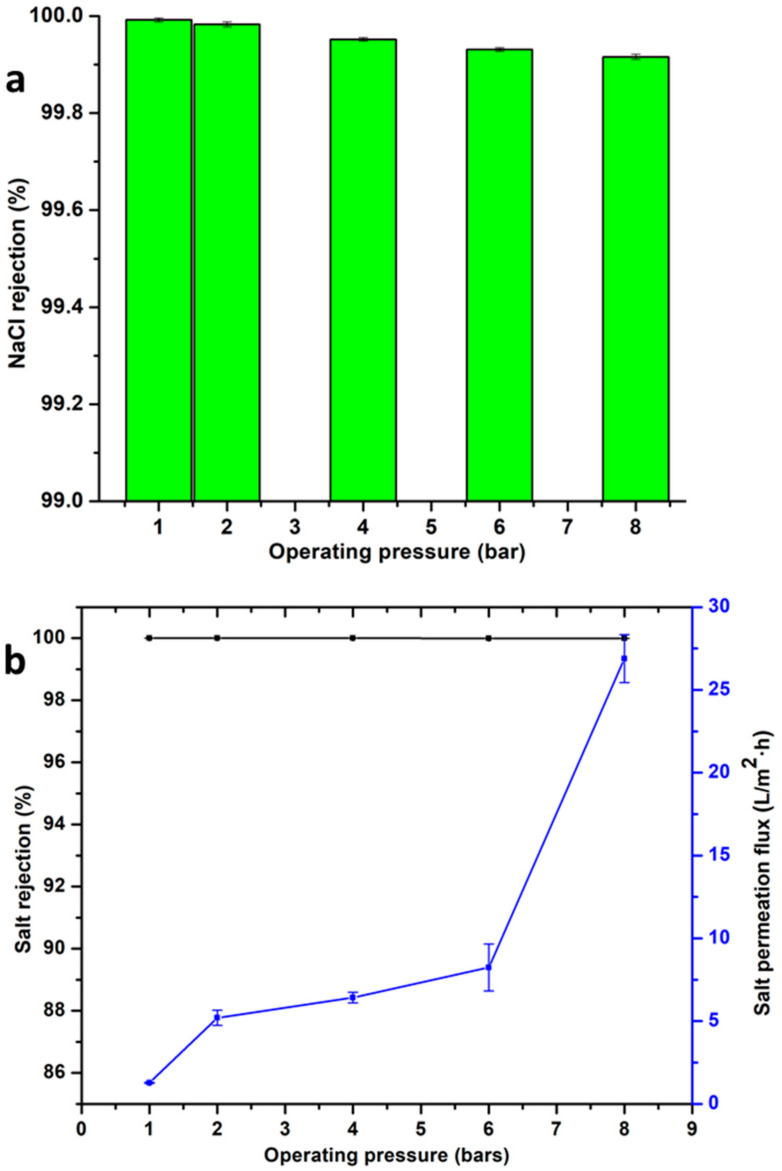
Salt rejection and salt permeation fluxes for the X48h membrane as a function of operating pressure (**a**) NaCl rejections, and (**b**) the synthetic seawater desalination and salt permeation fluxes in response to increasing operating pressures.

**Figure 8 membranes-10-00439-f008:**
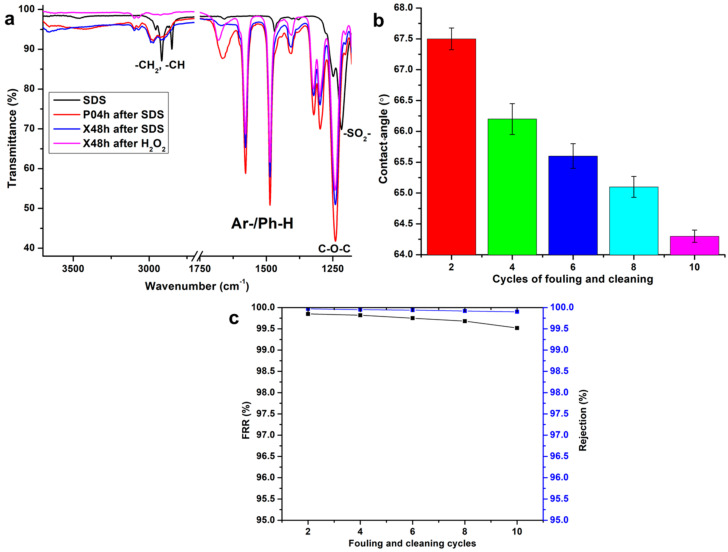
Testing of membrane integrity after fouling and cleaning: (**a**) ATR-FTIR spectra for SDS and the X04h, X08h, X24h, and X48h membranes; (**b**) Contact angle for the X48h membrane after fouling and cleaning; and (**c**) flux recovery ratios (black lines) and synthetic seawater rejection (blue lines) for the X48h membrane.

**Table 1 membranes-10-00439-t001:** Membrane variants according to the drying times.

Long Name	Notation
Pristine PES-air 4 h	P04h
xGnP-DA/PES-air 4 h	X04h
xGnP-DA/PES-air 8 h	X08h
xGnP-DA/PES-air 24 h	X24h
xGnP-DA/PES-air 48 h	X48h

**Table 2 membranes-10-00439-t002:** The composition character of the NaCl and synthetic seawater solutions used for performance determination.

Salts (m/v%)						
NaCl	MgCl_2_	Na_2_SO_4_	CaCl_2_	KCl	NaHCO_3_	Total
35.01	-	-	-	-	-	35.01
24.54	11.1	4.09	1.16	0.69	0.20	40.62

**Table 3 membranes-10-00439-t003:** The physiological character of the NaCl solution and synthetic seawater.

Solution	Conductivity (mS)	Total Dissolved Solids (TDS, ppt)	pH	Temperature (°)
Synthetic seawater	47.7	95.5	6.63	20.1
NaCl	4.29	2.12	5.57	22.5

**Table 4 membranes-10-00439-t004:** The XPS core levels, binding energies, and surface chemical composition/coverage and peak area measurements of the pristine PES and fabricated xGnP-DA/PES-air membranes.

	Pristine PES			xGnP-DA/PES-Air		
Core-Level	BE (eV)	Peak Area (cps)	Surface Composition (%)	BE (eV)	Peak Area (cps)	Surface Coverage (%)
C 1s	284.6	4833	50.3	284.6	2507.6	40.3
	285.6	1863.4	19.4	285.6	2929.6	52.2
	287.2	2197.9	30.3	287.2	611.1	9.82
				288.7	175.8	2.82
N 1s	-	-	-	398.4	487	13.5
				399.9	2173	60.6

**Table 5 membranes-10-00439-t005:** Pore sizes and contact angle values for the fabricated membranes.

Membrane	Pore size	Contact Angle (°)
P04h	0.21 µm	66.2 ± 0.2
X04h	0.17 µm	68.1 ± 0.3
X08h	0.12 µm	71.3 ± 0.5
X24h	0.09 µm	70.2 ± 0.8
X48h	0.00028 µm	67.5 ± 0.4

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
