# Peer review of "A New Method for a Polyethersulfone-Based Dopamine-Graphene (xGnP-DA/PES) Nanocomposite Membrane in Low/Ultra-Low Pressure Reverse Osmosis (L/ULPRO) Desalination"

_membranes, 2020, doi:10.3390/membranes10120439_

Round 1

Reviewer 1 Report

In this review, the authors prepared a mixed-matrix membrane incorporated with graphene-based filler to prepare dense membranes suitable for desalination. The discussions are interesting and the manuscript is very readable with good amount of evidence. I recommend publication after the authors addressed the following concerns:

1. Housekeeping: The standard of english is good with only minor spelling mistakes. Please kindly proofread again. 

2. The authors provided a overview of some of the desired properties of graphene in the introduction. While they are not wrong, some of the properties mentioned are irrelevant to the mixed matrix membrane design reported in this work such as crosslinking for stability is not going to be of help given that the graphene sheets are already embedded within the polymer matrix.  I recommend that authors should focus on properties such as tuning of the interlayer spacing by crosslinkers, the intrinsic impermeability of the graphene surface, leading to changes in the tortuosity of the transport pathways as some of the highlights worthy of discussion. The authors can refer to the following paper to deepen the discussion: https://doi.org/10.1016/j.carbon.2016.08.077

3. The percentage loading of xGnP-DA is not clear from the materials and methods. Can the authors please do the calculation and add in the weight percent of the filler in the mixed-matrix membrane for the reader?

4. Figure 1: The transmission is not very meaningful for FTIR spectrum. Can the authors please perform an offset so that the spectra are spread out from one another. I am unable to clearly identify the absorption bands.

5. For which membrane did the authors performed XPS on?

6. The SEM results from Figure 4 do not match with the conclusion gotten from FTIR. The authors claim that the slow upward movement of the xGnP-DA resulted in low intensity of the IR band which is in contrary to the SEM images shown in c1 and d1. If the worm-like structure is as claimed to be from the heightened presence of xGnP-DA/PES segment towards and on the membrane surface, then why is it that the FTIR bands are less intense for X48h?

7. I suggest to use point symbol for Figure 5. Easier for the reader to read.

8. Pore size in Table 5 should keep to only one type of units so that the reader can compare the magnitude of the values. 

9. The authors claimed that there is no significant effect on salt retention due to increased operational pressure, but in the following statement they claimed that there is no tradeoff as commonly reported. So the question is do the authors consider the change significant or not significant?

10. The water permeability increase with a tradeoff in salt rejection as the operating pressure increase is in contrary to what was reported in the literature. Typically, at higher pressure, membrane compaction and increase concentration polarization would have reduced water permeability and salt rejection. Can the author explain this?

11. Overall, in this entire paper, the role of XGnP-DA is not clear. Did the graphene derivative work as a barrier to increase the dense layer on the membrane surface or does it work as additive to tune the membrane structure? It appears that XGnP-DA serves more like an additive to give a dense selective layer and antifouling properties as evidenced by the separation and fouling data. In this regard, are there any value of using the graphene derivative in this work?

Author Response

Dear Reviewer

Please find the responses to your review comments in the PDF attached.

Regards

Authors

Reviewer 2 Report

The author reported the new fabrication method of ultra-low reverse osmosis (ULPRO) membranes using dopamine and graphene. The obtained results are good; however, they are a little doubtful. The evaluation method of the membrane performance seems to be not suitable. All the figure caption are not kindful and lack the detailed information. The detailed comments are as follows.

Page 3, Line 130
The information of the AFM manufacture and cantilever are lacked. Was the measurement carried out in air?

Page 4, Line 142
The author described that Jw was measured by using dead-end cell. Generally, Jw in the RO process is evaluated by cross-flow cell to prevent the change of the concentration of the feed water and the increase of osmotic pressure. If by using dead-end cell, the applied pressure will continue increasing. Why did the author use the dead-end cell? How to controll the applied pressure? Did the author consider the effect of the concentration polarization?

Page 4, Line 149
The author used the Guerout-Elford-Ferry equation to calculate the pore sizes. Is it appropriate to the sub-nano pores?

Page 12, Line 339 (Table 5)
The pore size dramatically changed from X24h to X48h. The author should evaluate this tendency between 24 and 48 more carefully. Additionally, the value of X48h is much smaller than that of polyamide RO. Is it reasonable value? It should be verify, for example, by measuring the rejection for not the synthetic wastewater but NaCl solution, because it is a surprising value.

Page 14, Line 378 (Figure 6)
I can not distinguish the flux values of X04h, X08h, X24h, and X48h, although they should be focused as ULPRO membranes. P04h should be shown separately because it is the support membrane. What is the mean of the inserted figure? Pressure vs flux?

Page 15, Line 408 (Figure 7)
Is this figure for X48h? The values of salt rejection are extremely high. Is it reasonable value? How is the NaCl rejection?

Author Response

Dear Reviewer

Please find the point-by-point responses to the review attached.

Regards

Authors

Round 2

Reviewer 1 Report

I have previously reviewed the manuscript and the authors have appropriately addressed my concerns. I recommend publication of the manuscript as it is. 

Author Response

Thank you for the review. The comments and the suggested corrections were very helpful in improving the manuscript.

Reviewer 2 Report

The author revised the manuscript to response my comments; however, didn't response against some comments properly.

Whole the manuscript
I have pointed out in first review that all the figure caption are not kindful and lack the detailed information. I can see no improvement. Ex. Which membranes did evaluate in Figure 7 and 8(b) and (c)? No explanation.

Page 4, Line 147
All of the obtained rejection is extremely higher than commercial polyamide RO membranes evaluated via a cross-flow method. That's why I can not believe these membrane performance results. The author should show the validity of this evaluation method; for example by measuring and comparing commercial RO membranes. If these values are correct, then, the author should discuss about the position of the membrane performance in the other reported membranes and commercial membranes, and why the fabricated membranes showed such high rejection.

Page 4, Line 157
I have pointed out the suitability of the Guerout-Elford-Ferry equation for the calculation of the sub-nano pore size of RO membranes. The author responded there is not enough literature for sub-nano pores. The author should comment this fact and the suitability of this equation to sub-nano pores, at least should refer some literature. The presented manuscript has only self-citation.

Author Response

Please find the responses attached.

Round 3

Reviewer 2 Report

The author responded to my comment properly, and I agree to the acceptance of this paper.